# Acute severe asthma requiring invasive mechanical ventilation in the era of modern resuscitation techniques: A 10-year bicentric retrospective study

Antoine Binachon[1,2], Adeline Grateau[1], Nicolas Allou[1], Cyril Ferdynus[3,4], Jérôme Allyn[1], Laurence Dangers[1], Olivier Martinet[1], Véronique Boisson[2], Alexandre Gauthier[2], Julien Jabot[1], Romain Persichini🆔[1]*

1 Médical–Surgical Intensive Care Unit, CHU Felix-Guyon, Saint-Denis, La Réunion, France, 2 Medical–Surgical Intensive Care Unit, CHU Sud-Réunion, Saint-Pierre, La Réunion, France, 3 Methodological Support and Biostatistics Unit, CHU Felix-Guyon, Saint-Denis, La Réunion, France, 4 INSERM, CIC 1410, Saint-Pierre, La Réunion France

* romain.persichini@hotmail.fr

**Data Availability Statement:** All relevant data are within the manuscript and its Supporting Information files.

## Abstract

### Purpose

Patients with acute severe asthma (ASA) may in rare cases require invasive mechanical ventilation (IMV). However, recent data on this issue are lacking.

### Materials and methods

In this retrospective and bicentric study conducted on a 10 year period, we investigate the in-hospital mortality in patients with ASA requiring IMV. We compare this mortality to that of patients with other types of respiratory distress using a standardized mortality ratio (SMR) model.

### Results

Eighty-one episodes of ASA requiring IMV were evaluated. Factors significantly associated with in-hospital mortality were cardiac arrest on day of admission, cardiac arrest as the reason for intubation, absence of decompensation risk factors, need for renal replacement therapy on day of admission, and intubation in pre-hospital setting. Non-survivors had higher SAPS II, SOFA, creatinine and lactate levels as well as lower blood pressure, pH, and $HCO_3$ on day of admission. In-hospital mortality was 15% (n = 12). Compared to a reference population of 2,670 patients, the SMR relative to the SAPS II was very low at 0.48 (95% CI, 0.25–0.84). The only factor independently associated with in-hospital mortality was cardiac arrest on day of admission. In-hospital mortality was 69% in patients with cardiac arrest on day of admission and 4% in others (p < 0.01). Salvage therapies were given to 7 patients, sometimes in combination with each other: ECMO (n = 6), halogenated gas (n = 1) and anti-IL5 antibody (n = 1). Death occurred in only 2 of these 7 patients, both of whom had cardiac arrest on day of admission.

**Funding:** The author(s) received no specific funding for this work.

**Competing interests:** The authors have declared that no competing interests exist.

## Conclusion

Nowadays, the mortality of patients with ASA requiring IMV is low. Death is due to multi-organ failure, with cardiac arrest on day of admission being the most important risk factor. In patients who did not have cardiac arrest on day of admission the mortality is even lower (4%) which allows an aggressive management.

## Introduction

Asthma is a chronic inflammatory disease of the respiratory tract that affects 235 to 300 million people worldwide and that is responsible for approximately 180,000 deaths annually [1–3]. In France, asthma affects more than 4 million people (6–7% of the population), with approximately 1,000 deaths per year [4]. The worldwide mortality rate of asthma does not seem to decrease over time [5]. Moreover, the prevalence of asthma has been increasing in recent years, indicating that the disease remains a serious global health problem [1, 2]. The prevalence of asthma may even increase in the future with increasing population migration around the world [6].

Acute severe asthma (ASA) is an asthma attack that does not respond to usual treatment and that can potentially lead to respiratory distress and death. However, there is no clear and consensual clinical definition of ASA at present [7–9]. Accordingly, recent French recommendations provide a pragmatic definition of ASA as any asthma exacerbation that is life-threatening or requires emergency management or both [10].

Patients with ASA may in rare cases (approximately 1% to 4%) require invasive mechanical ventilation (IMV) [11–13]. Indications for intubation and IMV are not well defined, but include respiratory or cardiac arrest, coma and extreme exhaustion, but also hypercapnia with severe acidosis despite optimal treatment [14, 15]. The mortality of patients with ASA requiring IMV varies considerably between studies. In a large review of studies conducted between 1977 and 2001, the mortality rate ranged from 0% to 38%, with a mean mortality rate of 8% [7]. More recent studies found mortality rates ranging from 0% to 8% [16–20]. The reasons for these discrepancies are not clear. One reason may be the lack of patient homogeneity, with some studies recruiting all patients with ASA hospitalized in the intensive care unit (ICU) [7, 17, 19], some studies including patients undergoing either invasive or non-invasive ventilation [13, 20], and finally others including only intubated patients [16, 18]. Moreover, most of the studies examined in the review by McFadden et al. are small, with an average of 40 patients per study [7]. Finally, some of the studies reporting a high mortality rate are old, which suggests that recruited patients may not have benefitted from all of the modern techniques that are available today. Indeed, salvage therapies such as extracorporeal membrane oxygenation (ECMO) are now used to treat refractory respiratory distress, with promising results in the case of patients with ASA [21].

In view of these elements concerning patients with ASA requiring IMV in the era of modern resuscitation techniques, this retrospective and bicentric study has been implemented. The main objectives were to determine the in-hospital mortality and to investigate the factors associated with this in-hospital mortality. The secondary objective was to compare the mortality of these patients to that of patients with other types of respiratory distress encountered in the ICU. Finally, this study aimed to describe: (I) the main characteristics of these patients; (II) the therapeutic management at admission; (III) the use of salvage therapies such as ECMO; (IV) and the complications encountered in ICU.

## Materials and methods

### Study design and population

This study retrospectively analyzed patients hospitalized for ASA in 1 of the 2 ICUs of the French overseas territory of Reunion Island over a 10-year period (from January 2008 to December 2017). All hospitalized adult patients ($\geq$ 18-year-old) who received a primary diagnosis of ASA requiring IMV were included. The 2 ICUs involved in this study, the University Teaching Hospital of Saint-Denis and the University Teaching Hospital of Saint-Pierre, are the only healthcare facilities of Reunion Island (850,000 inhabitants according to the last census) where patients can receive prolonged IMV. Patients were admitted to ICU either directly from home by the pre-hospital medical team or from 1 of the 4 emergency departments of the island.

The electronic database of the 2 ICUs was reviewed to identify eligible patients. Medical records were then analyzed to confirm that patients had been well diagnosed with ASA. For instance, all records of patients requiring mechanical ventilation for exacerbation of chronic obstructive pulmonary disease or congestive heart failure were specifically excluded. Medical records were analyzed by 2 investigators (AB, AG) to ensure that patients satisfied these inclusion criteria. Disagreements between the 2 principal investigators were resolved after discussion with a third investigator (RP).

### Data collection

The following patient characteristics were recorded: age, gender, height, weight, smoking, alcohol and/or drug abuse, chronic diseases, duration of asthma, presence of atopy, asthma treatment, previous hospitalization(s) for ASA requiring IMV or not, and compliance to treatment. Decompensation risk factor(s), intubation management (location and indications), vital signs on admission (blood pressure, heart rate, pulse oxygen saturation, temperature) were also recorded. Laboratory results (including blood gas results) were recorded on admission to ICU. Severity and comorbidity scores (Simplified Acute Physiology Score II (SAPS II) and Sequential Organ Failure Assessment (SOFA)) on the day of admission were calculated. Occurrence of cardiac arrest on the day of admission was investigated carefully based on medical records.

Data on the medical management of patients on the day of admission were collected. These data included: medications used for sedation, hemodynamic support, specific asthma treatment (all types of bronchodilators, corticosteroids, neuromuscular blockade, ketamine, and intravenous magnesium sulfate), mechanical ventilator settings, fluid administration, and use of renal replacement therapy. Data on the use of various types of salvage therapies (ECMO, halogenated gas, immunotherapy, or any other unconventional treatment), including indications for treatment, were also gathered. Indications for ECMO support were decided on a case-by-case basis. Indeed, because ASA is a very rare indication for ECMO, there is no specific protocol for the administration of ECMO in patients with ASA, unlike what is the case for patients with refractory ARDS. It should be noted that a mobile ECMO team is available in Reunion Island. Finally, we recorded the complications encountered during ICU stay, namely: ICU-acquired weakness, barotrauma, ventilator-associated pneumonia or other ICU-acquired infections, shock related to ASA (cardiac tamponade due to gas trapping), severe atelectasis, myocardial ischemia, and severe cardiac arrhythmia. The rate of tracheotomy was also recorded.

## Ethical and legal considerations

According to French law, no informed consent is necessary for the extraction of anonymous data or for the anonymous analysis of patients' medical records. This study was approved by the Ethics Committee of the "Société de Réanimation de Langue Française" (# CE SRLF 18–10). This study was approved by the French National Commission on Informatics and Liberty (# 2184166v0). This study follows the STROBE statement recommendations (STrengthening the Reporting of OBservational studies in Epidemiology).

## Data analysis

Patients' characteristics were expressed as frequencies and proportions for categorical variables and as medians and interquartile ranges for quantitative variables. Bivariate comparisons were performed using Pearson's Chi-Square test or Fisher's exact test for qualitative data and Student's t-test or the Mann-Whitney test for quantitative data, as appropriate. Backward logistic regression was conducted to determine the factors associated with in-hospital mortality. All factors associated with a p-value < 0.20 in bivariate analysis were entered in the model and then removed if they did not reach the significance level of p < 0.05. Adjusted Odds Ratios and their 95% Confidence Intervals were estimated. All tests were performed with a 2-tailed type I error of 5%. All analyses were made using SAS 9.4 (SAS Institute Inc., Cary, NC).

A reference population was created to compare the mortality of ASA to that of other types of respiratory distress. The respiratory distress group was composed of patients with a primary diagnosis of hypoxemic respiratory distress, hypercapnic respiratory distress, acute respiratory distress syndrome, or ASA. Medical records were selected based on the International Classification of Diseases, 10th Edition, coding. Inclusion was limited to patients requiring IMV who were hospitalized during the same period (from January 2008 to December 2017) and in the same 2 Reunion Island ICUs as patients hospitalized for ASA. Finally, the observed mortality of ASA was compared to the expected mortality according to the reference population using a standardized mortality ratio (SMR). The SMR was calculated from SAPS II quartile of the reference population with any type of respiratory distress.

## Results

A total of 81 patients with ASA requiring IMV were admitted to 1 of the 2 Reunion Island ICUs between January 2008 and December 2017 (Fig 1). The estimated incidence of patients with ASA requiring IMV was 5 per 1,000 patients admitted to the 2 ICUs. Table 1 shows patient's general characteristics and history of asthma. The majority of the 81 patients were female (64%) and rather young (49 [42–60] years old). All patients were known to have asthma. Atopy was found in 87% of the patients, and the diagnosis of asthma was made 21 [13–31] years earlier. A quarter of the patients had a previous episode of ASA that required IMV. Usual treatments for asthma were mainly short-acting beta-agonists (91%) and inhaled corticosteroids (73%). However, 27% (n = 22) of the patients had no treatment for asthma. The main comorbidities (which were often multiple) were smoking, diabetes, chronic alcohol abuse, chronic obstructive pulmonary disease, immunosuppressive therapy (including systemic corticosteroids), and chronic renal failure. The majority of patients had no comorbidities (58%).

### Characteristics of patients at admission and ASA risk factors

In almost half of the cases, no decompensation risk factors were found. Identified decompensation risk factors were: bacterial or viral pneumonia (15%), acute bronchitis (14%), cessation

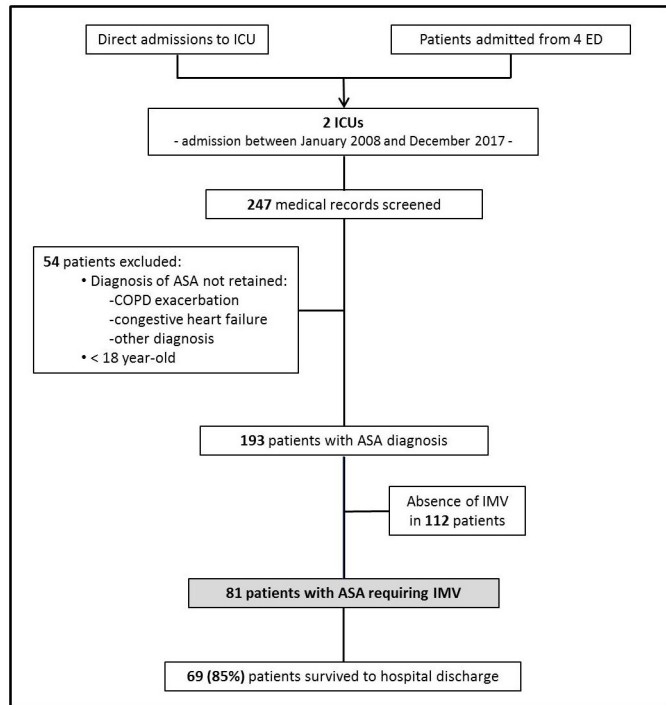

**Fig 1. Flowchart of patient selection.** ICU: intensive care unit; ED: emergency department; ASA: acute severe asthma; COPD: chronic obstructive pulmonary disease; IMV: invasive mechanical ventilation.

of asthma treatment (12%), exposure to allergens (10%), and non-respiratory sepsis (4%). The location of intubation was mainly the emergency department (54%), followed by the pre-hospital setting (24%) and the ICU (22%). Intubation was performed in 74 patients (91%) on day 1, in 6 patients (7%) on day 2, and in 1 patient (1%) on day 3. The main intubation criteria were: signs of respiratory exhaustion, hypercapnia, hypoxemia, altered state of consciousness, and cardiac arrest. The SAPS II was 40 [30–54] and the SOFA score was 5 [2–7]. Thirteen patients presented a cardiac arrest on the day of admission (6 patients in the pre-hospital setting, 6 in the emergency department, and 1 in the ICU). All other characteristics at admission and risk factors are presented in Table 2.

## Therapeutic management on the day of admission

The majority of the patients were treated with a short-acting beta-agonist (96%), often in association with a short-acting muscarinic antagonist (87%). Short-acting agonists were always administered with a nebulizer. Intravenous short-acting beta-agonists were given to only 8% of the patients. Systemic corticosteroids were administered to 83% of the patients. The majority of patients (65%) received continuous neuromuscular blockade on the day of admission. Nearly half of the patients (45%) had hemodynamic failure that required infusion of catecholamines (sometime in combination with each other) on the day of admission: norepinephrine was given to 39% of the patients, epinephrine to 23% of the patients, and dobutamine to none of the patients. To note, adrenalin was used when both Beta1 and Beta2 effects were needed (inotropic effect and bronchodilator effect). Only 5% of the patients needed renal replacement therapy on the day of admission. Patients received 1,000 [0–1,750] mL of fluids on the day of admission. Mechanical ventilator settings on the day of admission were as follows: a tidal volume of 405 [375–455] mL, a respiratory rate of 15 [12–16] breaths/min, a positive end-

**Table 1. Patients' general characteristics and history of asthma according to in-hospital survival status.**

| Variables | All patients (n = 81) | | Survivors (n = 69) | | Non-survivors (n = 12) | | p |
|---|---|---|---|---|---|---|---|
| | DA (n) | Median [IQR] or n (%) | DA (n) | Median [IQR] or n (%) | DA (n) | Median [IQR] or n (%) | |
| Age (years) | 81 | 49 [42–60] | 69 | 49 [42–57] | 12 | 57 [42–70] | 0.23 |
| Male gender | 81 | 29 (36%) | 69 | 28 (47%) | 12 | 1 (8%) | **0.05** |
| BMI (kg/m$^2$) | 59 | 24.9 [21.2–27.8] | 52 | 24.8 [21.2–27.7] | 7 | 27.0 [24.8–31.2] | 0.25 |
| Smoking | 80 | 33 (41%) | 68 | 29 (43%) | 12 | 4 (33%) | 0.75 |
| Alcoholism | 81 | 17 (21%) | 69 | 15 (22%) | 12 | 2 (17%) | 1.00 |
| Cannabis use | 81 | 8 (10%) | 69 | 7 (10%) | 12 | 1 (8%) | 1.00 |
| Comorbidities | | | | | | | |
| Cirrhosis | 81 | 2 (3%) | 69 | 1 (1%) | 12 | 1 (8%) | 0.28 |
| Diabetes | 81 | 21 (26%) | 69 | 17 (25%) | 12 | 4 (33%) | 0.50 |
| COPD | 81 | 8 (10%) | 69 | 8 (12%) | 12 | 0 (0%) | 0.60 |
| Chronic renal failure | 81 | 4 (5%) | 69 | 2 (3%) | 12 | 2 (17%) | 0.10 |
| Cancer | 81 | 1 (1%) | 69 | 1 (1%) | 12 | 0 (0%) | 1.00 |
| Immunosuppressive therapy | 81 | 5 (6%) | 69 | 3 (4%) | 12 | 2 (17%) | 0.16 |
| No comorbidities | 81 | 47 (58%) | 69 | 41 (59%) | 12 | 6 (50%) | 0.54 |
| Atopy | 38 | 33 (87%) | 33 | 28 (84.8%) | 5 | 5 (100%) | 1.00 |
| Onset of asthma symptoms (years) | 45 | 21 [13–31] | 38 | 20 [11–30] | 7 | 38 [14–46] | **0.04** |
| Compliance to usual asthma treatment | 69 | 38 (55%) | 60 | 34 (57%) | 9 | 4 (44%) | 0.72 |
| Previous hospitalization in ICU | 81 | 28 (35%) | 69 | 23 (33%) | 12 | 5 (42%) | 0.74 |
| with invasive mechanical ventilation | 81 | 20 (25%) | 69 | 18 (26%) | 12 | 2 (17%) | 0.72 |
| Usual asthma treatment | | | | | | | |
| Long-acting beta agonist | 81 | 56 (69%) | 69 | 50 (73%) | 12 | 6 (50%) | 0.17 |
| Short-acting beta agonist | 81 | 74 (91%) | 69 | 63 (91%) | 12 | 11 (92%) | 1.00 |
| Long-acting muscarinic antagonist | 81 | 14 (17%) | 69 | 13 (19%) | 12 | 1 (8%) | 0.68 |
| Inhaled corticosteroid | 81 | 59 (73%) | 69 | 52 (75%) | 12 | 7 (58%) | 0.29 |
| Systemic corticosteroid | 81 | 12 (15%) | 69 | 10 (15%) | 12 | 2 (17%) | 1.00 |
| Leukotriene-Receptor Antagonist | 81 | 19 (24%) | 69 | 17 (25%) | 12 | 2 (17%) | 0.72 |
| Anti-IgE therapy | 81 | 8 (10%) | 69 | 7 (10%) | 12 | 1 (8%) | 1.00 |

DA: data available; IQR: interquartile range; BMI: body mass index; COPD: chronic obstructive pulmonary disease; ICU: intensive care unit.

expiratory pressure of 2 [0–5] cmH$_2$O, and a FiO$_2$ of 60 [50–100] %. These results are summarized in S1 Table (Supplemental Digital Content).

## Salvage therapies

Salvage therapies were given to 7 patients, sometimes in combination with each other: ECMO was given to 6 patients, halogenated gas to 1 patient, and anti-IL5 antibody (mepolizumab) to 1 patient. The characteristics and outcome of these 7 patients are shown in S2 Table.

## Outcome

Nearly half of the patients experienced complications: 25% of the patients had ventilator-associated pneumonia, 15% had other ICU-acquired infections, 7% had severe atelectasis requiring specific management, 6% had ICU-acquired weakness, 6% had severe cardiac arrhythmia, 6% had myocardial ischemia, 6% had shock related to ASA, and 5% had barotrauma (pneumothorax, pneumomediastinum, or subcutaneous emphysema). The rate of tracheostomy was 7%.

**Table 2. Characteristics of patients and organ failure on the day of admission according to in-hospital survival status.**

| Variables | All patients (n = 81) | | Survivors (n = 69) | | Non-survivors (n = 12) | | p |
|---|---|---|---|---|---|---|---|
| | DA (n) | Median [IQR] or n (%) | DA (n) | Median [IQR] or n (%) | DA (n) | Median [IQR] or n (%) | |
| Decompensation risk factor | | | | | | | |
| Non-compliance to treatment | 81 | 10 (12%) | 69 | 10 (15%) | 12 | 0 (0%) | 0.34 |
| Exposure to allergens | 81 | 8 (10%) | 69 | 8 (12%) | 12 | 0 (0%) | 0.60 |
| Pneumonia (viral or bacterial) | 81 | 12 (15%) | 69 | 9 (13%) | 12 | 3 (25%) | 0.37 |
| Bronchitis | 81 | 11 (14%) | 69 | 11 (16%) | 12 | 0 (0%) | 0.20 |
| Non-respiratory sepsis | 81 | 3 (4%) | 69 | 3 (4%) | 12 | 0 (0%) | 1.00 |
| Unknown | 81 | 40 (49%) | 69 | 31 (45%) | 12 | 9 (75%) | **0.05** |
| Location of intubation | | | | | | | |
| Pre-hospital setting | 81 | 19 (24%) | 69 | 12 (17%) | 12 | 7 (58%) | **0.05** |
| ED | 81 | 44 (54%) | 69 | 41 (59%) | 12 | 3 (25%) | **0.03** |
| ICU | 81 | 18 (22%) | 69 | 16 (23%) | 12 | 2 (17%) | 1.00 |
| Intubation criteria | | | | | | | |
| Cardiac arrest | 81 | 11 (14%) | 69 | 4 (6%) | 12 | 7 (58%) | **<0.01** |
| Shock | 81 | 5 (6%) | 69 | 3 (4%) | 12 | 2 (17%) | 0.16 |
| Sign of respiratory exhaustion | 81 | 54 (67%) | 69 | 49 (71%) | 12 | 5 (42%) | 0.09 |
| Hypoxemia | 81 | 24 (30%) | 69 | 22 (32%) | 12 | 2 (17%) | 0.49 |
| Hypercapnia | 81 | 41 (51%) | 69 | 37 (54%) | 12 | 4 (33%) | 0.19 |
| Coma | 81 | 20 (25%) | 69 | 19 (28%) | 12 | 1 (8%) | 0.28 |
| Vital signs on admission to ICU | | | | | | | |
| SAP (mmHg) | 78 | 122 [104–154] | 68 | 124 [106–157] | 10 | 102 [90–124] | **0.01** |
| DAP (mmHg) | 78 | 69 [60–82] | 68 | 69 [61–85] | 10 | 60 [57–70] | **0.03** |
| Heart rate (/min) | 80 | 113 [100–123] | 69 | 115 [102–124] | 11 | 100 [90–117] | 0.07 |
| Saturation (%) | 80 | 99 [96–100] | 69 | 99 [97–100] | 11 | 98 [94–100] | 0.54 |
| Temperature (°C) | 79 | 36.1 [35.4–36.8] | 68 | 36.1 [35.6–36.8] | 11 | 35.4 [33.4–36.5] | **0.05** |
| Biology on admission to ICU | | | | | | | |
| Platelets (G/L) | 78 | 251 [200–297] | 67 | 249 [205–298] | 11 | 259 [157–273] | 0.69 |
| Total bilirubin (mmol/L) | 64 | 6.0 [4.5–10.5] | 56 | 6.0 [5.0–10.5] | 8 | 5.0 [4.0–10.0] | 0.62 |
| Creatinine (μmol/L) | 80 | 79.5 [66.0–105.0] | 69 | 76.0 [65.0–92.0] | 11 | 121.0 [90.0–148.0] | **<0.01** |
| PaO$_2$ (mmHg) | 80 | 153 [100–233] | 69 | 148 [102–230] | 11 | 163 [82–299] | 0.82 |
| PaCO$_2$ (mmHg) | 80 | 66.5 [54.0–86.0] | 69 | 66.0 [54.0–87.0] | 11 | 72.0 [54.0–85.0] | 0.68 |
| pH | 80 | 7.16 [7.08–7.25] | 69 | 7.18 [7.11–7.25] | 11 | 7.01 [6.84–7.10] | **<0.01** |
| Lactate (mmol/L) | 78 | 2.3 [1.5–4.0] | 68 | 2.2 [1.3–3.5] | 10 | 9.3 [2.9–15.0] | **<0.01** |
| HCO$_3^-$ (mmol/L) | 80 | 24.5 [21.0–29.0] | 69 | 25.0 [21.6–29.0] | 11 | 21.3 [15.0–26.0] | **0.03** |
| PaO$_2$/FiO$_2$ ratio | 67 | 296 [170–370] | 58 | 296 [184–370] | 9 | 269 [108–299] | 0.25 |
| SAPS II | 81 | 40 [30–54] | 69 | 39 [28–50] | 12 | 66 [54–79] | **<0.01** |
| SOFA | 81 | 5 [2–7] | 69 | 4 [2–7] | 12 | 10 [6–11] | **<0.01** |
| Cardiac arrest on day of admission (pre-hospital setting, ED or ICU) | 81 | 13 (16%) | 69 | 4 (6%) | 12 | 9 (75%) | **<0.01** |

DA: data available; IQR: interquartile range; ED: emergency department; ICU: intensive care unit; SAP: systolic arterial pressure; DAP: diastolic arterial pressure; SAPS: simplified acute physiology score; SOFA: sequential organ failure assessment.

Hospital and ICU length of stay was 13 [8–17] and 7 [4–12] days, respectively, with IMV being administered for a period of 4 [2–9] days. These data are summarized in S3 Table.

In-ICU mortality was 14% (n = 11) and in-hospital mortality was 15% (n = 12). One death occurred in the medical ward after ICU discharge due to hypoxic cardiac arrest caused by a

**Table 3. Standardized mortality ratio of patients with acute severe asthma relative to the SAPSII compared to a reference population of patients with any type of respiratory distress.**

| SAPS II per quartile | Nb of patients in the Respiratory Distress cohort | Mortality in the Respiratory Distress cohort | Nb of patients in the ASA cohort | Expected Nb of deaths in the ASA cohort | Observed Nb of deaths in the ASA cohort |
|---|---|---|---|---|---|
| < 39 | 644 | 19.6% | 33 | 6.46 | 2 |
| 39–51 | 677 | 25.8% | 22 | 5.69 | 0 |
| 52–66 | 661 | 38.1% | 15 | 5.72 | 4 |
| > 66 | 688 | 64.8% | 11 | 7.13 | 6 |
| Total | 2,670 | 37.4% | 81 | 25 | 12 |

Calculation of the **Standardized Mortality Ratio** = (12/25) = **0.48 (95%CI, 0.25–0.84)**.

SAPS II quartiles were defined according to a reference population of 2,670 patients with any type of respiratory distress requiring IMV, who were hospitalized during the same period (from January 2008 to December 2017) and in the same 2 ICUs as patients hospitalized for ASA.

The respiratory distress group was defined by a primary diagnosis of hypoxemic respiratory distress, hypercapnic respiratory distress, acute respiratory distress syndrome or ASA.

SAPS: simplified acute physiology score; ASA: acute severe asthma.

tracheostomy mucus plug. The characteristics and in-hospital evolution of the 12 non-surviving patients are presented in S4 Table.

A total of 2,670 patients with any type of respiratory distress requiring IMV were identified to create the reference population. Table 3 presents the SMR of patients with ASA for each quartile of SAPS II of the reference population. Compared to this population, the SMR of patients with ASA relative to the SAPS II was very low at 0.48 (95%CI, 0.25–0.84).

## Factors associated with in-hospital mortality

Factors associated with in-hospital mortality in bivariate analysis are presented in Tables 1, 2, S1 and S3. Non-survivors were more likely to be women with longer duration of asthma. Cardiac arrest on the day of admission, cardiac arrest as the reason for intubation, need for renal replacement therapy on the day of admission, absence of decompensation risk factors, and intubation in the pre-hospital setting were significantly associated with in-hospital mortality. By contrast, intubation in the emergency department was associated with survival. Compared to survivors, non-survivors had higher SAPS II, SOFA, creatinine levels, and lactate levels as well as lower blood pressure, pH, and $HCO_3$ on the day of admission.

In multivariate analysis, the only factor independently associated with in-hospital mortality was cardiac arrest on the day of admission. In-hospital mortality was 69% (n = 9) in the 13 patients who presented with cardiac arrest on the day of admission and 4% (n = 3) in the other 68 patients (p < 0.01) (Tables 2 and S4).

## Discussion

Our results suggest that patients with ASA requiring IMV have low mortality. Indeed, the in-hospital mortality of patients recruited in our study was low at 15%, and was even lower (4%) for the sub-group of patients who did not present a cardiac arrest on the day of admission.

These results are in line with older studies, which reported mortality rates ranging from 0% to 38% for patients with ASA requiring IMV [7, 22]. Likewise, in the largest retrospective study of ASA in ICU, Gupta et al. found a global mortality rate of 9.8% for 2152 patients admitted to 128 ICUs of the United Kingdom between 1995 and 2001. However, the mortality rate rose to 15% when only intubated patients were considered [17]. Two recent studies which also focused on patients with ASA requiring IMV reported mortality rates of 0% and 3%,

respectively [16, 18]. These rates are lower than that observed in our study, but the populations in these 2 studies included pediatric patients and were therefore globally younger than our study population. Moreover, the majority of patients in these 2 studies had no effective treatment for chronic asthma: only 43% and 28% of patients were treated with inhaled corticosteroids [16, 18], which are currently the first-line treatment for asthma [2]. By contrast, 73% of our patients were treated with inhaled corticosteroids. The higher mortality observed in our study may be explained by the fact that our patients had more severe asthma, with decompensation occurring despite appropriate treatment. In the most recent study by Althoff et al., the in-hospital mortality rate of ASA patients requiring IMV was also low (2.4%). However, these patients were less severely ill with only a small proportion of patients under vasopressors (<11% vs 39% in our study). Moreover, the rate of cardiac arrest was not reported. [20]. In summary, the global in-hospital mortality of our cohort seems comparable with other recent studies when considering only severely ill intubated patients.

However, our in-hospital mortality results must be qualified. The in-hospital mortality rate of 15% was much lower than the in-hospital mortality rate of 25% predicted by the SAPS II score on admission (calculated at 40 [30–54]). Moreover, compared to patients with any kind of respiratory distress requiring IMV, the SMR of patients with ASA relative to the SAPS II was very low at 0.48, and was even lower in the lower SAPS II quartile (Table 3). These findings can be explained by the fact that the pathophysiology associated with ASA as a single disorder is fully (and sometimes quickly) reversible provided the appropriate treatment is applied. Moreover, patients with ASA were mostly young (49[42–60] years old) and free of comorbidities (58% in our cohort—Table 1), which reduced the risk of decompensation of underlying diseases.

Indeed, in our study, death was mainly due to multiple organ failure complicating cardiac arrest. As in other studies of ASA [17], cardiac arrest on the day of admission was by far the most important risk factor: mortality was 69% in patients who presented with cardiac arrest on the day of admission vs. 4% in other patients (p < 0.01), even in those who eventually required salvage therapies. These results are consistent with the study of Abdelkarim et al.; the in-hospital mortality of asthmatic patients under IMV was very low (1.1% to 2.4%), but rose to 57% in patients who suffered cardiac arrest [19]. In view of these findings, we recommend the administration of maximalist treatment to patients who do not present with cardiac arrest on the day of admission, with a fairly high probability of success even in the most critically ill patients.

Insofar as the complications that we have reported are concerned, an interesting point of our study is the higher than usually reported incidence of ventilator associated pneumonia [23]. Indeed, 25% of our patients presented with a ventilator-associated pneumonia which corresponds to 24 episodes per 1000 ventilator-days. In the study of Secombe et al., the incidence of pneumoniae was only 5%, but non-intubated asthmatic patients were included [24]. To our knowledge, there are no recent studies which report the incidence of ventilator-associated pneumonia in the specific case of ASA patients requiring IMV. Comparison with non-asthmatic ventilated patients is therefore difficult. Among the hypotheses that could be evoked to explain the higher incidence in ASA patients requiring IMV, one could consider the role of early corticosteroid therapy and / or underlying bronchial pathology as an area for further studies to clarify this discrepancy. As far as other common complications are concerned (pneumothorax, pneumomediastinum, ICU-acquired weakness, myocardial ischemia) the incidence in our study was generally 1.5 to 2 times higher than reported by Secombe et al. in a wide cohort of intubated and non-intubated patients [24]. This could also be explained by the highest severity of asthmatic patients under IMV compared to non-intubated asthmatic patients.

Our study also explored the efficacy of salvage therapies (S2 Table). It has been suggested that halogenated gas ensures adequate ventilation by decreasing airway resistance [25]. While the one patient who was treated with halogenated gas in our study had a good outcome, the evidence for the efficacy of this technique is based only on small case reports [26]. Indications for this treatment, which is not included in recent recommendation [10], have yet to be clearly defined. The same applies to immunotherapy targeting hypereosinophilic asthma. While the latter seems promising for the treatment of severe chronic asthma [27, 28], no data are available on its use in ICU except in case-reports [29]. We used this medication for a single patient only along with eosinophilic bronchoalveolar lavage, based on expert advice. This patient, who was also treated with venovenous ECMO, had a good outcome. It should be noted that for the treatment of severe chronic asthma, the positive effects of immunotherapy seem to appear after several weeks of treatment [28]. It is thus difficult to attribute the favorable outcome of this patient to the use of immunotherapy alone. As a consequence, the use of immunotherapy should not be recommended for ASA patients requiring IMV and further studies are needed to establish the indication of this kind of treatment in ICU. By contrast, several reviews exist on the use of ECMO in patients with ASA refractory to usual care. According to the review by Hebbar et al. [30], ECMO has been used successfully in children with ASA refractory to usual care, with a 94% rate of survival. In adults, the study by Mikkelsen et al., which also relied on the ELSO registry (Extracorporeal Life Support Organization) found an in-hospital survival rate of 83% in a cohort of 24 adults treated with ECMO for ASA between 1986 and 2006 [31]. For patients treated with ECMO for other indications (pneumonia, pulmonary embolism, pulmonary hypertension, trauma) the in-hospital survival rate was only 51% [31]. The relationship of ASA as the indication for ECMO initiation with a better outcome remained significant after adjustment for each potentially confounding variable (age, pre-ECMO cardiac arrest, race, gender, duration of mechanical ventilation prior to ECMO initiation, etc.) [31]. More recently, Schmidt et al. found a survival rate of 94% for 35 patients with ASA treated with venovenous ECMO between 2000 and 2012 [32]. For other indications, the survival rate ranged from 45% to 70% [32]. In the latter study ASA as the indication for ECMO initiation was the factor associated with better outcome, with an OR of 17.7. Consequently, it was the parameter associated with the best survival rate as per the RESP-score, which was developed to predict survival in patients receiving venovenous ECMO for any type of respiratory distress [32, 33]. The recent study by Yeo et al., involving 272 patients treated with ECMO for ASA between 1992 and 2016, confirms the good outcome for this indication. In the study, the weaning success rate was 87% and the in-hospital survival rate was 84% [34]. It should be noted that this last study highlighted a non-negligible risk of hemorrhagic complications directly responsible for death in approximately 1.5% of patients [34]. Our study confirms these findings. Indeed, death occurred in only 2 of the 6 patients who received ECMO, both of whom presented a cardiac arrest on the day of admission and received venoarterial ECMO for cardiogenic shock (S2 Table). These 2 patients died not directly from refractory ASA, but as a result of their cardiac arrest (brain death and multiple organ failure with acute mesenteric ischemia–S4 Table). The 4 other patients treated with ECMO who did not present a cardiac arrest on the day of admission survived to hospital discharge (S2 Table). In view of these good results, venovenous ECMO should be considered as an effective tool for highly-selected ASA patients refractory to usual care [21]. However, precise clinical indications for ECMO have yet to be specified [10].

The main strength of our study is the comprehensiveness of the data. The latter applied to all of Reunion Island, since the only 2 ICUs of this French overseas territory were involved in the study. In other words, all Reunionese patients with ASA requiring IMV during the study period were analyzed.

However, our study also has limitations that must be acknowledged. The first limitation is the retrospective nature of the study. Consequently, data collection is unfortunately sometimes not exhaustive: lack of anthropometric data, lack of respiratory mechanics values (plateau pressure, driving-pressure, intrinsic PEEP, I:E ratio), etc. In addition, the selection of patients may have resulted in bias. Second, one might argue that our results cannot be transposed to other regions. Indeed, it has been suggested that the prevalence of asthma is higher in Reunion Island than in mainland France [4]. However, a recent study has cast doubt on this hypothesis: it found that the prevalence of asthma among adults in Reunion Island is identical to that in mainland France [35], which itself corresponds to the mean of other developed countries [36]. Third, data collection was spread over a period of 10 years, raising the possibility that changes occurred in the global management of asthma. Yet, no major advances have been made in the treatment of asthma during the study period, and all patients in our study received treatments based on recent recommendations [10]. Fourth, our results concerning the SMR relative to SAPS II have to be interpreted carefully. Indeed, SAPS II does not take into account the history of chronic pathologies, unlike other ICU scores such as Charlson scores and APACHE II. Unfortunately, only SAPS II is collected prospectively on the medical records in our units, and this is the reason why it was chosen to calculate the SMR of our patients to avoid biases. Results could have been different with the use of other scoring methods. Finally, the main limitation concerns the use of salvage therapies. The good results obtained in our cohort in well-selected patients (i.e. those who have not had cardiac arrest on the day of admission) cannot be generalized. Indeed, the low number of salvage therapies reported here does not allow us to prove that their use has a causal link with the survival of the patients in whom it has been used. In conclusion, even if these results are very encouraging, these salvage therapies must remain exceptional treatments which should be reserved for highly selected patients.

## Conclusion

Our study shows that nowadays, the mortality of patients with ASA requiring IMV is lower than that of patients with any other type of respiratory distress encountered in the ICU. Death was only due to multi-organ failure, with cardiac arrest on the day of admission being the most important risk factor. In patients who did not present with cardiac arrest on the day of admission, mortality was even lower. In view of these findings, patients with ASA who do not present a cardiac arrest on the day of admission should be given aggressive treatment, including salvage therapies like ECMO. However, other studies are needed to specify the precise indications and the selection of patients for such exceptional treatments.

## Supporting information

**S1 Table. Therapeutic management on the day of admission and use of salvage therapies according to in-hospital survival status.**
(DOCX)

**S2 Table. Characteristics and outcome of the 7 patients who received salvage therapies.**
(DOCX)

**S3 Table. Complications and outcome according to in-hospital survival status.**
(DOCX)

**S4 Table. Characteristics of the 12 patients who died in hospital according to whether or not they presented a cardiac arrest on the day of admission.**
(DOCX)

## Acknowledgments

We are grateful to Arianne Dorval and Andrew Hobson for English proofreading of the manuscript and to Michel Bohrer for the exploitation of the database.

## Author Contributions

**Conceptualization:** Antoine Binachon, Julien Jabot, Romain Persichini.

**Formal analysis:** Cyril Ferdynus, Romain Persichini.

**Investigation:** Antoine Binachon, Adeline Grateau, Julien Jabot, Romain Persichini.

**Methodology:** Antoine Binachon, Adeline Grateau, Nicolas Allou, Cyril Ferdynus, Jérôme Allyn, Laurence Dangers, Olivier Martinet, Véronique Boisson, Alexandre Gauthier, Julien Jabot, Romain Persichini.

**Software:** Cyril Ferdynus.

**Supervision:** Julien Jabot, Romain Persichini.

**Writing – original draft:** Antoine Binachon, Adeline Grateau, Romain Persichini.

**Writing – review & editing:** Nicolas Allou, Cyril Ferdynus, Jérôme Allyn, Laurence Dangers, Olivier Martinet, Véronique Boisson, Alexandre Gauthier, Romain Persichini.

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
