## [Decision Letter · Decision Letter 0]

18 Aug 2020

PONE-D-20-20581

Acute severe asthma requiring invasive mechanical ventilation in the era of modern resuscitation techniques: a 10-year bicentric retrospective study

PLOS ONE

Dear Dr. PERSICHINI,

Thank you for submitting your manuscript to PLOS ONE. After careful consideration, we feel that it has merit but does not fully meet PLOS ONE’s publication criteria as it currently stands. Therefore, we invite you to submit a revised version of the manuscript that addresses the points raised during the review process.

We look forward to receiving your revised manuscript.

Kind regards,

Corstiaan den Uil

Academic Editor

PLOS ONE

Journal Requirements:

Reviewers' comments:

Reviewer's Responses to Questions

**Comments to the Author**

1. Is the manuscript technically sound, and do the data support the conclusions?

Reviewer #1: Yes

2. Has the statistical analysis been performed appropriately and rigorously? 

Reviewer #1: Yes

3. Have the authors made all data underlying the findings in their manuscript fully available?

Reviewer #1: Yes

4. Is the manuscript presented in an intelligible fashion and written in standard English?

Reviewer #1: Yes

5. Review Comments to the Author

Reviewer #1: This paper summarises the retrospective experience of ventilated asthma from an electronic database over 10 years. As such it contributes to a small number of patients reported in the literature.

1. The strengths of this paper is the report of ECMO outcomes and greater detail of the international experience of ECMO in asthma should be reported in the discussion.

2. There was a high incidence of ventilator associated pneumonia which seems at variance with other papers - could this be expanded upon and how that diagnosis was made

3. A number pf patients received immunotherapy - could the rationale for this be expanded upon as it is unlikely to benefit in acute therapy

4. The use of other respiratory failure seems an arbitrary basis by which to report a SMR in the asthma patients - the details of the matching needs to be described as SAPS II would seem not to be a good way to do this for asthma especially in that asthma patients have few comorbidities and perhaps would be better done with a charlson score as well or perhaps APACHE in include the chronic health points in SAPS II are very limited and severe in nature and there is a need to ensure that the respiratory failure is not compared in surgical or trauma patients- so much greater detail is needed on this area and whether this approach is a validated way to do it.

5. There are more recent reviews of asthma managed in the ICU which the present data could be compared to

6. In how many were as the IV adrenaline given for blood pressure support or for asthma? Was the Magnesium given as an infusion? Were bronchodilators given by MDI or nebuliser? At which point were the ventilator settings to report selected?

7. Was there any information available on the degree of asthma control prior to admission and what follow-up was organised for these patients?

6. PLOS authors have the option to publish the peer review history of their article (what does this mean?). If published, this will include your full peer review and any attached files.

Reviewer #1: **Yes: **Robert Boots

---

## [Author Response · Author response to Decision Letter 0]

16 Sep 2020

Response to Reviewers

We thank the reviewer for his comments, all of which have been considered as follows. 

1. The strengths of this paper is the report of ECMO outcomes and greater detail of the international experience of ECMO in asthma should be reported in the discussion.

We thank the reviewer for this suggestion. 

More details on the experience of ECMO in asthma have been added in the discussion: other interesting results on the 2 studies of Mikkelsen and Schmidt have been added, and in addition, we now mention the recent study of Yeo et al. (Extracorporeal membrane oxygenation for life-threatening asthma refractory to mechanical ventilation: analysis of the Extracorporeal Life Support Organization registry, Crit Care 2017) [page 18 – lines 336 to 355].

2. There was a high incidence of ventilator associated pneumonia which seems at variance with other papers - could this be expanded upon and how that diagnosis was made.

We agree with the reviewer on this point. In our study, the crude incidence rate is not especially higher than what we find in the most recent literature (Koulenti et al., Eur J Clin Microbiol Infect Dis 2017), but as the average length of stay is short in our study (7 [4-12] days), the incidence rate per 1000 days is actually higher at 24 episodes per 1000 ventilator-days . This point is now discussed in the revised manuscript in a specific paragraph on complications [page 16 – lines 302 to 312].

According to international and French recommendations, diagnosis was made using a combination of clinical respiratory deterioration, a modification of the Chest x-ray and by the presence of a positive bacterial culture (quantitative or semi-quantitative) on a respiratory sample.

3. A number of patients received immunotherapy - could the rationale for this be expanded upon as it is unlikely to benefit in acute therapy.

We fully agree with the reviewer that this kind of treatment is not common in ICU. This type of medication is typically indicated in chronic hypereosinophilic asthma. 

Only one patient received immunotherapy (mepolizumab) in our study.

Doubts and questions concerning the use of this type of treatment in acute therapy have been emphasized in the revised manuscript [page 17 – lines 325 to 333].

4. The use of other respiratory failure seems an arbitrary basis by which to report a SMR in the asthma patients - the details of the matching needs to be described as SAPS II would seem not to be a good way to do this for asthma especially in that asthma patients have few comorbidities and perhaps would be better done with a Charlson score as well or perhaps APACHE in include the chronic health points in SAPS II are very limited and severe in nature and there is a need to ensure that the respiratory failure is not compared in surgical or trauma patients- so much greater detail is needed on this area and whether this approach is a validated way to do it.

We agree with the reviewer on this point and we thank him for this valuable remark. SAPS II indeed does not take account of the history of chronic pathologies, unlike Charlson scores and APACHE II. Unfortunately, only SAPS II is collected prospectively on the medical records in our units, which is why it was chosen to construct the Standardized Mortality Ratio of our patients. If we had tried to find the APACHE and Charlson scores retrospectively by searching the medical records of the patients, this would have resulted in significant biases due to too many missing data.

However, taking into account this important limitation, it should be noted that this method has already been used in 3 other published studies by our team on ARDS and/or septic shock:

-Mortality of critically ill patients with severe influenza starting four years after the 2009 pandemic, David Vandroux et al., Infect Dis (Lond) 2019.

-Acute respiratory distress syndrome in leptospirosis, David Vandroux et al., J Crit Care 2019.

-Leptospirosis in ICU: A Retrospective Study of 134 Consecutive Admissions, Benjamin Delmas et al., Crit Care Med 2018).

This important point is now discussed as a limitation of the study in the discussion section of the revised manuscript [pages 19/20 – lines 380 to 386].

5. There are more recent reviews of asthma managed in the ICU which the present data could be compared to.

This is a very pertinent comment indeed. As proposed by the reviewer, we reviewed again the most recent literature. We found four other interesting reviews and studies that we have added to the bibliography.

The main results of these studies have been added to the discussion and provides more perspective and context:

-A comparison of characteristics and outcomes of patients admitted to the ICU with asthma in Australia and New Zealand and United states, Abdelkarim et al., J Asthma 2020 [citation 19 and discussion page 16 – lines 296 to 298].

-Noninvasive Ventilation Use in Critically Ill Patients with Acute Asthma Exacerbations, Althoff et al., Am J Respir Care Med 2020 [citation 20 and discussion page 15 – lines 275 to 279].

-Clinical management practices of life-threatening asthma: an audit of practices in intensive care, Secombe et al., Crit Care Resusc 2019 [citation 24 and discussion pages 16/17 – lines 305 to 315].

-Utilization of Mechanical Ventilation for Asthma Exacerbations: Analysis of a National Database, Nanchal et al., Respir Care 2014 [citation 13].

6. In how many were as the IV adrenaline given for blood pressure support or for asthma? Was the Magnesium given as an infusion? Were bronchodilators given by MDI or nebuliser? At which point were the ventilator settings to report selected?

Indeed, those are important matters on which we need to be more specific:

-IV adrenaline: as the data were collected retrospectively, we cannot be absolutely sure of the indication the practitioner in charge had in mind. However, in our clinical practice, drugs indications are fairly standardized:

° we use IV salbutamol when we are looking for a bronchodilator effect only.

° when we are looking for a pressor effect only, we use norepinephrine. 

° when we are looking for an inotropic effect only, we use dobutamine.

° therefore, adrenalin is used most often for its double Beta1 and Beta2 effects, i.e. as an inotropic drug for cardiogenic shock associated with acute severe asthma but also for its bronchodilator effect.

-Magnesium: always given as an infusion.

-Bronchodilators: given by nebulizer only (no MDI devices are used in our ICU).

-Ventilator settings: the reported ventilator settings were reported by the practitioner in charge on the first prescription sheet at day one, just after patient admission and clinical evaluation at the bedside.

The sentences concerning these treatments and the ventilator settings have been modified; we hope that it is now clearer [page 11 – lines 200 to 214].

7. Was there any information available on the degree of asthma control prior to admission and what follow-up was organized for these patients?

Unfortunately, we do not have accurate information about asthma control of our patients prior to admission.

It must be recognized that the degree of asthma control (shortness of breath, use of rescue inhaler, nocturnal awakening, etc.) is unfortunately often poorly researched with relatives when patients are admitted to intensive care. This notion may appear, wrongly, to be secondary to the immediate seriousness of the patient.

The only information we have collected that could approach this notion is the history of previous ICU admissions (Table 1). 

In the majority of cases, the patients were transferred from ICU to the pneumology ward where follow up took place before hospital discharge. Unfortunately, we do not have access to this specific data.

We hope that we provided a satisfactory answer to all the reviewer’s criticisms and requests and that the manuscript is significantly improved.

Respectfully yours,

Romain PERSICHINI on behalf of all the co-authors.

---

## [Editor Report · Decision Letter 1]

18 Sep 2020

Acute severe asthma requiring invasive mechanical ventilation in the era of modern resuscitation techniques: a 10-year bicentric retrospective study

PONE-D-20-20581R1

Dear Dr. PERSICHINI,

We’re pleased to inform you that your manuscript has been judged scientifically suitable for publication and will be formally accepted for publication once it meets all outstanding technical requirements.

Kind regards,

Corstiaan den Uil

Academic Editor

PLOS ONE
---

## [Editor Report · Acceptance letter]

23 Sep 2020

PONE-D-20-20581R1 

Acute severe asthma requiring invasive mechanical ventilation in the era of modern resuscitation techniques: a 10-year bicentric retrospective study 

Dear Dr. Persichini:

I'm pleased to inform you that your manuscript has been deemed suitable for publication in PLOS ONE. Congratulations! Your manuscript is now with our production department. 

Kind regards, 

on behalf of

Dr. Corstiaan den Uil 

Academic Editor

PLOS ONE